# Long-term visual prognosis and characteristics of recurrent retinal detachment after silicone oil removal

**Wookyung Park[1,2], Mirinae Kim[1,2], Rae Young Kim[1,2], Joo Young Kim[1,2], Jae Hyuck Kwak[1,2], Young-Gun Park[1,2], Young-Hoon Park ![ORCID][1,2]***

**1** Department of Ophthalmology and Visual Science, College of Medicine, The Catholic University of Korea, Seoul, Korea, **2** Catholic Institute for Visual Science, College of Medicine, The Catholic University of Korea, Seoul, Korea

* parkyh@catholic.ac.k

**Data Availability Statement:** Data cannot be shared publicly because of Possibility of personal information leakage. Data are available from the

## Abstract

### Purpose

Silicone oil (SO) is commonly used for tamponade purposes in retinal detachment (RD) surgery, but the long-term visual prognosis after removal of the oil, and in particular, what is known about the recurrence of RD after SO removal, remains unclear. The purpose of this study is to evaluate the long-term vision prognosis after SO removal, and to understand the frequency and characteristics of RD recurrence.

### Methods

We retrospectively reviewed the medical charts of 1017 eyes of patients with a diagnosis of RD who had a pars plana vitrectomy with SO tamponade between January 2009 and December 2018. Best-corrected visual acuity (BCVA) was obatained before and after vitrectomy and also at the last visit. After SO removal, the group who showed improvement in visual acuity and the group who did not were compared. The anatomical results were compared between the group in which the retina was detached again after SO removal and the group in which the retina was not detached. To determine whether the duration of SO tamponade affects RD recurrence, further analysis was performed by dividing subgroups according to SO tamponade duration. RD recurrence, visual acuity, SO tamponade period were investigated.

### Results

Mean follow-up period was 56.65 ± 72.02 months. An average SO tamponade period was 6.68 ± 11.39 months. The average logMAR BCVA was 1.75 ± 0.91 before SO injection, 1.60 ± 0.75 before SO removal and 1.29 ± 0.96 after the removal. After SO removal, 926 of the 1017 (91.1%) patients had well attached retina without recurrence. There was no significant difference in visual acuity before SO removal in re-detachment group compared to no re-detachment group, but visual acuity of re-detachment group was worse than no re-detachment group after SO removal (p<0.001). The SO tamponade period in the group with

Institutional Review Board of the Catholic University of Korea ('seoulirb@cmcnu.or.kr' and 'seoul_irb@catholic.ac.kr') for researchers who meet the criteria for access to confidential data.

**Funding:** The author(s) received no specific funding for this work.

**Competing interests:** The authors have declared that no competing interests exist.

improved vision after SO removal was 5.09 ± 9.87 months, and the period was significantly shorter than the 9.09 ± 13.05 months in the group not showing vision recovery (p = 0.005). The occurrence of corneal opacity was significantly higher in the group with SO over 6 months, than those of the two groups with SO tamponade duration of less than 3 months and between 3 and 6 months (p = 0.038). The longest tamponade group showed the worst final vision after SO removal (p<0.001).

## Conclusion

The prognosis for final vision is generally good when performing surgery using SO in RD, but considering the complications that arise after surgery, long-term retention of SO is not recommended and the timing of SO removal should be considered.

## Introduction

Rhegmatogenous retinal detachment (RRD) is the most common form of retinal detachment (RD) that an ophthalmologist can meet in clinical practice [1]. RRD is one of the emergency diseases in which the ophthalmic prognosis such as vision can rapidly deteriorate when treatment is delayed. In addition to RRD, tractional retinal detachment (TRD) caused by diabetes can also cause severe and rapid loss of vision, and blindness can occur if surgery is not performed [2]. The treatment options include primary scleral buckle, pars plana vitrectomy with tamponade (silicone oil, gas) or a combination of both [3]. Prolonged silicone oil (SO) tamponade can cause complications like cataract, elevated intraocular pressure, hypotony, emulsification and keratopathy [4, 5]. Therefore, it has been suggested that SO should be removed 3 to 6 months after putting it in the eye [6, 7]. Retinal re-detachment is a major complication after removal of SO which ranges from 2% to 33% of cases according to several studies [5, 8]. One of the suggested mechanisms of retinal re-detachment is traction that occurs or worsens after removal of SO in a portion that has been suppressed with the oil [9, 10]. The eyes that had SO removal are filled with water instead of vitreous. Compared to viscous vitreous, body fluids are more likely to enter tiny retinal holes and cause RD recurrence.

SO removal is somewhat common surgical procedure in several vitreoretinal diseases, but it is still unclear in terms of long-term visual prognosis after a removal of the oil. Meanwhile not so much of characteristics of re-detachment of retina is unfolded yet. The purpose of this study is to evaluate the long-term visual prognosis and determine the frequency and characteristics of retinal re-detachment after SO removal in RD patients.

## Patients and methods

This study was carried out retrospectively and followed the Declaration of Helsinki. An approval was obtained from the Institutional Review Board of the Catholic University of Korea. Considering the aspect of retrospective study, necessity of obtaining informed patient consent was waived.

All patients were recruited between January 2009 and December 2018 at Seoul St. Mary's Hospital in Korea. We retrospectively reviewed the medical charts of 1017 eyes of patients with a diagnosis of RD who had a pars plana vitrectomy (ppV) with SO tamponade. RD cases such as RRD, tractional retinal detachment (TRD) and proliferative vitreoretinopathy (PVR) were included. RD cases included both primary cases that had not previously been operated

on, and cases that had recurred after surgery at this institution or at another hospital. All the macula or the optic nerve related diseases such as macular hole or epiretinal membrane that were confirmed before RD diagnosis were excluded.

Intraocular surgery was performed by five experienced vitreoretinal surgeons and consisted of a 23- or 25- gauge ppV (Alcon Constellation Vision System; Alcon Laboratories Inc.). Perfluorocarbonliquid (Bausch and Lomb, USA), intravitreal triamcinolone and indocyanine green (Dongindang pharmaceutical, South Korea) were used at surgeon's discretion. The internal limiting membrane (ILM) was peeled in the presence of PVR at the discretion of the surgeon. SO was used for tamponade (Arcadophta, France).

BCVA was measured before SO injection surgery and 1 month after the surgery. BCVA was also measured when hospitalized for SO removal surgery, and was measured at 1 month, 3 months, 6 months, and 1 year after removal, respectively. BCVA was measured at the last visit for patients who did not visit until 1 year after surgery and patients who had follow up for more than 1 year. BCVA was assessed using Snellen charts. Either Swept source Optical Coherence Tomography (OCT) (Topcon, Japan) or spectral-domain OCT (Heidelberg, Germany), wide angle fundus photography (Zeiss, Germany) and ocular ultrasound (Zeiss, Germany) were performed before and after the surgery.

Outcome measures were BCVA, retinal adhesion and unwanted complications such as secondary glaucoma, corneal opacity, and phthisis bulbi. Statistical analysis was carried out using SPSS version 24.0 (SPSS Inc, Chicago, Illinois, USA). BCVA was converted into logarithm of the minimum angle resolution (LogMAR) VA for analysis. All descriptive data is presented as medians and ranges. The Mann–Whitney U test was used to compare the predefined outcome measurements. Differences with $p < 0.05$ were considered to be statistically significant. To analyze the groups divided by the duration of the SO tamponade, ANOVA test and Kruskal-Wallis test were performed on continuous variables, and Chi-squared test and Fisher's exact test were performed on categorical variables. As it concerns exploratory data analysis, no adjustments for multiple testing were performed.

## Results

A total of 1017 patients were included in the study. 370 eyes (36.4%) underwent concomitant scleral buckling surgery at the time of retinal surgery. Mean follow-up period were $56.65 \pm 72.02$ months. And the SO was removed after stabilizing the retina in the eyes for an average of $6.68 \pm 11.39$ months. The retina remained attached after removal of silicone oil in 926 of the 1017 (91.1%) patients included in the study. The most frequent indication of surgery was RRD (722 eyes, 71.0%) and TRD as a complication of progressive diabetic retinopathy, was the second most frequent cause of surgery. Overall, visual acuity improved after removal of oil than before surgery (Table 1).

In the comparison of the no re-detachment group and the re-detachment group after the oil removal surgery, there was no difference in age and sex between the two groups. The duration of SO tamponade did not show any significant difference between groups either ($6.57 \pm 9.89$ vs. $7.74 \pm 21.70$, $p = 0.612$) (Table 2). Before the SO removal surgery, visual acuity has not shown significant difference between the no re-detachment and re-detachment groups ($1.60 \pm 0.75$ vs. $1.54 \pm 0.81$, $p = 0.692$), but the visual acuity measured at the latest visit was significantly better in no re-detachment group ($1.21 \pm 0.92$ vs. $2.12 \pm 0.91$, $p<0.001$).

For PVR, which may have an important effect on post-operative prognosis, such as recurrence of RD or visual acuity, there was no significant difference between the two groups ($p = 0.917$). In addition to TRD due to PDR, diabetes mellitus could adversely affect retinal disease but there was no difference in prevalence of the disease between the two groups

**Table 1. Demographics, surgical and visual outcomes (n = 1017).**

| | |
|---|---|
| Age (years) | 51.89 ± 15.59 |
| Sex (Male, %) | 614 (60.4%) |
| Concomitant buckling (Yes, %) | 370 (36.4%) |
| Previous retinal surgeries | |
| Once | 715 (70.3%) |
| Twice | 218 (21.4%) |
| More than 3 times | 84 (8.3%) |
| Presence of PVR (Yes, %) | 158 (15.5%) |
| Presence of Diabetes (Yes, %) | 322 (31.7%) |
| SO tamponade period (months) | 6.68 ± 11.39 |
| Mean F/U period (months) | 56.65 ± 72.02 |
| Indications for silicone oil | |
| RRD (%) | 722 (71.0%) |
| TRD (%) | 218 (21.4%) |
| MH RD (%) | 77 (7.6%) |
| Anatomical outcomes | |
| No re-detachment after SO removal | 926 (91.1%) |
| Redetachment after SO removal | 91 (8.9%) |
| BCVA (logMAR) | |
| Before SO inj | 1.75 ± 0.91 |
| Before SO removal | 1.60 ± 0.75 |
| Last F/U | 1.29 ± 0.96 |

Values are presented as n (%), mean ± standard errors.

PVR, Proliferative vitreoretinopathy; SO, silicone oil; F/U, follow up; RRD, Rhegmatogenous Retinal detachment; TRD, Tractional Retinal detachment; MH, macular hole; BCVA, best-corrected visual acuity

($p = 0.500$). And also the incidence of post-operative complications such as secondary glaucoma, corneal opacity, or phthisis bulbi was not significantly different between the two groups. The presence of scleral buckling or the number of mean previous retinal surgeries were not different between the no re-detachment group and the re-detachment group after removal of SO.

In the group with improved visual acuity after the removal of SO, there were no statistically significant differences in the presence of accompanying buckle, PVR or DM compared with decreased or stationary visual acuity group. The duration of the SO tamponade was 5.09 ± 9.87 months in the improved visual acuity group and 9.09 ± 13.05 months in the group not showing vision recovery, indicating that the lower visual acuity group had significantly longer SO ($p = 0.005$). The RRD and TRD, which accounted for the majority of the causes of RD, did not differ between the two groups, but the macular hole RD was higher in the group with reduced vision ($p = 0.030$). Corneal opacity, which had not existed before retinal surgery, was confirmed in 7 patients in the improved visual acuity group and 34 patients in the other. There was a significant difference between the two groups ($p = 0.002$). (Table 3).

To compare the effects of SO tamponade period, we compared patients divided into three" groups: patients who had SO under 3 months, 3 to 6 months and over 6 months (Table 4). Re-detachment rate was not different among the three groups ($p = 0.712$). Among complications that occurred after SO removal, secondary glaucoma did not differ between groups ($p = 0.348$), but corneal opacity was significantly higher in the group who had SO for 6 months or longer ($p = 0.038$). The visual acuity before SO removal did not differ between the

**Table 2. Analysis of anatomic outcomes.**

| | No re-detachment after SO removal (n = 926) | Redetachment after SO removal (n = 91) | P value |
|---|---|---|---|
| Age (years) | 51.97 ± 15.68 | 51.07 ± 14.89 | 0.776 |
| Sex (Male, %) | 561 (60.6%) | 54 (59.3%) | 0.900 |
| Concomitant buckling (Yes, %) | 336 (36.3%) | 34 (37.0%) | 0.934 |
| Mean previous retinal surgeries | 1.40 ± 0.76 | 1.59 ± 0.75 | 0.205 |
| Presence of PVR | 144 (15.6%) | 13 (14.8%) | 0.917 |
| Presence of DM | 299 (32.3%) | 23 (25.3%) | 0.500 |
| SO tamponade period (months) | 6.57 ± 9.89 | 7.74 ± 21.70 | 0.612 |
| Indications for silicone oil | | | |
| RRD (%) | 657 (71.0%) | 65 (71.4%) | 0.563 |
| TRD (%) | 195 (21.0%) | 23 (25.3%) | 0.553 |
| MH RD (%) | 74 (8.0%) | 3 (3.3%) | 0.424 |
| BCVA (logMAR) | | | |
| Before SO inj | 1.73 ± 0.92 | 1.93 ± 0.88 | 0.279 |
| Before SO removal | 1.60 ± 0.75 | 1.54 ± 0.81 | 0.692 |
| Last F/U | 1.21 ± 0.92 | 2.12 ± 0.91 | <0.001* |
| Complications | | | |
| Secondary glaucoma | 81 (8.7%) | 2 (2.2%) | 0.278 |
| Corneal opacity | 34 (3.6%) | 6 (6.6%) | 0.150 |
| Phthisis bulbi | 3 (0.4%) | 0 (0%) | 0.773 |

Values are presented as n (%), mean ± standard errors.

*P <0.05

BCVA, best-corrected visual acuity; SO, silicone oil; F/U, follow up; PVR, proliferative vitreoretinopathy; DM, Diabetes mellitus

**Table 3. Analysis of visual outcomes.**

| | BCVA improved (n = 614) | BCVA deteriorated/stationary (n = 403) | P value |
|---|---|---|---|
| BCVA (logMAR) | | | |
| Before SO inj | 1.57 ± 0.90 | 2.02 ± 0.87 | <0.001* |
| Before SO removal | 1.56 ± 0.71 | 1.65 ± 0.80 | 0.284 |
| Last F/U | 0.71 ± 0.47 | 2.18 ± 0.82 | <0.001* |
| Concomitant buckling (Yes, %) | 205 (33.4%) | 164 (40.7%) | 0.184 |
| Mean previous retinal surgeries | 1.37 ± 0.75 | 1.48 ± 0.77 | 0.210 |
| Presence of PVR | 94 (15.3%) | 64 (15.9%) | 0.900 |
| Presence of DM | 205 (33.4%) | 117 (29.0%) | 0.446 |
| SO tamponade period (months) | 5.09 ± 9.87 | 9.09 ± 13.05 | 0.005* |
| Indications for silicone oil | | | |
| RRD (%) | 440 (71.6%) | 282 (69.9%) | 0.317 |
| TRD (%) | 144 (23.5%) | 74 (18.4%) | 0.284 |
| MH RD (%) | 30 (4.9%) | 47 (11.7%) | 0.030* |
| Complications | | | |
| Secondary glaucoma | 37 (6.0%) | 47 (11.7%) | 0.080 |
| Corneal opacity | 7 (1.1%) | 34 (8.4%) | 0.002* |

Values are presented as n (%), mean ± standard errors.

*P <0.05

BCVA, best-corrected visual acuity; SO, silicone oil; F/U, follow up; PVR, proliferative vitreoretinopathy; DM, Diabetes mellitus; RRD, Rhegmatogenous Retinal detachment; TRD, Tractional Retinal detachment; MH, macular hole

**Table 4. Comparison of results according to silicone oil tamponade period.**

|  | Under 3months SO tamponade (n = 272) | 3–6 months SO tamponade (n = 460) | Over 6months SO tamponade (n = 285) | P value |
|---|---|---|---|---|
| Redetachment (%) | 23 (8.5%) | 47 (10.2%) | 20 (7.1%) | 0.712 |
| Mean previous retinal surgeries | 1.31 ± 0.58 | 1.33 ± 0.69 | 1.66 ± 0.95 | 0.002* |
| BCVA (logMAR) |  |  |  |  |
| Before SO inj | 1.60 ± 0.97 | 1.70 ± 0.89 | 1.98 ± 0.86 | 0.018* |
| Before SO removal | 1.65 ± 0.80 | 1.49 ± 0.69 | 1.72 ± 0.78 | 0.063 |
| Last F/U | 1.10 ± 0.90 | 1.13 ± 0.87 | 1.75 ± 0.99 | <0.001* |
| Complications |  |  |  |  |
| Secondary glaucoma | 30 (11.1%) | 34 (7.3%) | 20 (7.1%) | 0.348 |
| Corneal opacity | 10 (3.7%) | 10 (2.2%) | 30 (10.6%) | 0.038* |

Values are presented as n (%), mean ± standard errors.

*$P < 0.05$

BCVA, best-corrected visual acuity; SO, silicone oil; F/U, follow up

three groups, but after removal, the longest tamponade group showed statistically significant difference compared to the under 3 months tamponade group (p<0.001) and the 3 to 6 months group (p<0.001) respectively, and the visual acuity was the worst (Fig 1).

## Discussion

Clinically, SO removal is considered when the retina is judged sufficiently stable after SO tamponade. However, there are cases where the retina separates after removal. In these cases, the prognosis is usually poor even after retinal adhesions by reoperation. We tried to find out if there are any factors that significantly influence the recurrence of retinal detachment after SO removal.

In other similar studies, patient follow up was usually 6 months to 2 years [11–15]. We collected data of 1017 patients over 6 years after the SO removal. Although not divided according to the cause of the surgery, the group with reduced or stationary vision showed significantly longer SO tamponade period compared to the group with improved vision after SO removal (p = 0.005).

When the group was divided and analyzed for each period of SO tamponade, there was no difference in retinal detachment recurrence rate according to the tamponade period. Several previous studies have investigated whether there is a difference in SO tamponade duration between the group with recurrence of RD after SO removal and the group with immediate retinal adhesion [9, 16–18]. Overall, they showed that the attached eyes without relapse had a longer SO tamponade period than the relapsed eyes. The criteria for dividing groups were different for each study, so there may be some difficulties in comparison with this study, these studies did not perform subgroup analysis around 3 or 6 months as in this study.

Although there was no significant difference in visual acuity prior to removal of SO, after the removal the visual acuity was worse in the group who had SO for a period of 3 to 6 months than in the group with SO removed within 3 months, and the worst in the group who had had SO for 6 months or longer. Prior to SO injection, there was a significant difference in visual acuity between groups, which can be thought to indicate that there may have been an anatomical or functional difference in the state of the retina at the time of the first surgery, but since there is no adjustment for variables other than the tamponade period, it cannot be concluded

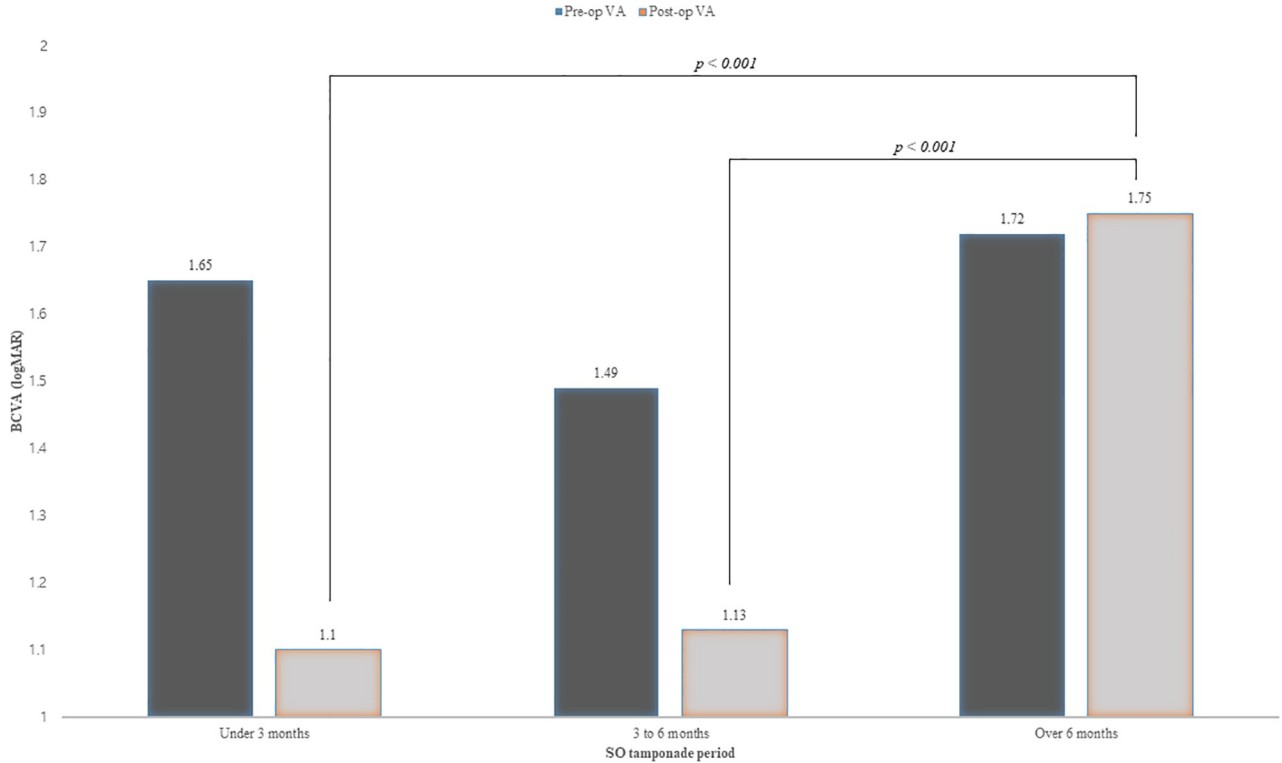

**Fig 1. Comparison of BCVA(logMAR) before and after SO removal according to SO tamponade period.** Prior to SO removal, there was no significant difference in BCVA between the three groups, but after removal, there was a significant difference in visual acuity between 3–6 months tamponade group and over 6 months tamponade group, and under 3 months tamponade group and over 6 months tamponade group.

prematurely. In addition, there was no difference in the incidence of secondary glaucoma, but the occurrence of corneal opacity was significantly higher in the group who had SO for more than 6 months (p = 0.038).

We investigated how PVR affects RD recurrence. In the present study, frequency of retinal re-detachment after removal of SO was 8.9%. Patients with PVR and RD had relapses in 8.2% after SO removal. Several studies have shown that PVR affects RD recurrence after SO removal [1, 3, 7–10, 19]. Abrahams et al. reported that retinal re-detachment was found 19% of their study patients, and 14% of them had a PVR [19]. He et al. found 18.6% of patients who had surgeries got re-detached retina, and among them, 15% of patients had PVR. However, even in the no re-detachment group, 13.7% of PVRs were identified, so it was difficult to say that PVR itself increases the risk of RD recurrence [10]. Although there are differences in reported results between studies, PVR has been accepted as an important factor in RD recurrence. The residual or newly formed traction after SO removal causes the retina to detach again, which is thought to occur due to the remaining vitreous base or fibrotic change due to retinal damage. Depending on the eye's condition, the surgical proficiency of the operator, the degree to which the PVR has been sufficiently removed, whether the band has been raised well and if the SRF has been sufficiently removed, the result may be different for each patients and thus there would be a limit in the analysis.

Although trying to analyze the effect of buckle band use, it is not possible to ethically divide patients into accompanying band group or ppV only group for research purposes, so the use of buckle placement in the study was not randomized between patients. If it is judged that the

retinal condition is bad, it can be sufficiently estimated that buckling surgery is performed together. Without a buckle band in the eye, even a small amount of traction may occur on the remains of the vitreous base near the equator, which can cause retinal re-detachment. In particular, when the retina around the retinal tear is no longer in direct contact with the vitreous base, it becomes difficult to exert direct traction on the retina. Jonas et al. reported that in the RD with PVR, the group that put the band together had a significantly lower rate of retinal re-detachment than the eyes that did not put the band [20]. Deaner et al showed that the RD patients who used ppV, retinectomy, SO tamponade, and without using buckle band showed the same or better surgical results compared to the results of previous studies in which buckle bands were applied under the same conditions [21]. As such, the conclusions of each study are different, and the result of our study was the percentage of concomitant buckling did not differ significantly between the "no re-detachment" and "re-detachment" groups. Garweg et al suggested that removing enough cytokines, RPE, and inflammatory cells during vitrectomy may have an advantage over buckling in RD surgery [22]. Diabetes, the degree of retinal detachment, and PVR grading cannot be divided, making it difficult to derive significant differences. Considering the pathology due to inflammation of PVR, it is thought that it is not considered sufficient to reduce recurrence by reducing the distance of physical traction by buckling itself.

The number of previous retinal surgeries done before SO injection was not significantly related to the occurrence of a retinal re-detachment after SO removal. Whether additional retinal surgery is required clinically depends on a number of factors, including the degree of PVR, the preoperative eye condition, and the presence of accompanying DM. Several studies found that the number of previous retinal surgeries were associated with the risk of retinal re-detachment after the SO removal in contrast to our study result [10, 20]. In the future, if variables other than the number of previous retinal surgery are matched and a follow-up study is conducted to compare the recurrence of RD, more accurate information will be obtained.

Several other studies have revealed that the surgeon factor is one of the main characteristics affecting postoperative anatomical success [5–7, 23–25]. Probably in most areas of clinical surgical field today, the same equipment, instruments and similar surgical techniques were used, and the surgical experience and results of five surgeons in this study were very similar. In a further study, we also could find out the difference from a surgeon factor meanwhile environmental condition should be maintained the same.

With a minimal time period between ppV and removal of SO of 2 month (usually longer than 5 months), duration of intraocular SO tamponade had no significant (p = 0.612) effect on the rate of postoperative retinal re-detachment. It suggests that the retinal situation may be settled within a not so long time. It may also be true in view of histological findings showing microscopic SO related changes in the retina occurring after 4 weeks of SO tamponade [26].

There are limitations of the present study. Since it is a clinical study on the outcome of a surgical procedure, there are numerous factors influencing the results. They may lead to somewhat insignificant difference and may mask significant differences between study groups. To reduce the influence of external factors, the study included only patients in whom ppV and removal of SO had been performed by same operation room setting including vitrectomy arrangements. An additional limitation of this study is the fact that visual acuity before SO injection was significantly different in the three groups of different SO tamponade period and this could have affected the visual prognosis after surgery. One of the limitations of this study is that the last follow-up time varies from patient to patient. Other factors may have played a role, especially with longer follow-up periods. Another limitation of a study like the present one is that the rate of postoperative retinal re-detachments is markedly influenced by the criteria to perform ppV and the criteria to use SO. It can make it difficult to compare studies on the same topic. The main goals of the present study were, however, not the assessment of the rate

of retinal re-detachments after removal of intraocular SO. The main purpose was to evaluate the risk factors leading to, and to examine the factors indicating, retinal re-detachment. These figures, compared with the overall rate of retinal re-detachment, may be less influenced by the criteria to use and to remove SO. In this study, it was confirmed that the duration of SO tamponade had no significant difference in retinal detachment and had an effect on postoperative visual acuity. Therefore, if it is judged that the retina is stable, it is not necessary to extend the duration of SO tamponade. A further important limitation of the study is that it is a retrospective non-comparative case series investigation, so that the patients were not randomly distributed between the various parameters tested in the study. It limits the generalizability of the results. For some variables, however, the non-randomized study design served only to support the findings.

In conclusion, our study suggests that when discussing ppV with the patient, one should take into account the timing of the retinal surgery, preoperative visual acuity, general condition of the patient and the benefit a potentially successful surgery may have on the quality of life of the patient. After carefully checking the structure of the retina to the far periphery, if it is confirmed that it is stably attached, there is no reason to delay the removal of SO. It can adversely affect long-term vision prognosis.

## Supporting information

**S1 File.**
(XLSX)

## Author Contributions

**Conceptualization:** Wookyung Park, Young-Hoon Park.

**Data curation:** Wookyung Park, Mirinae Kim, Rae Young Kim, Joo Young Kim, Jae Hyuck Kwak, Young-Hoon Park.

**Formal analysis:** Wookyung Park, Joo Young Kim, Jae Hyuck Kwak, Young-Gun Park.

**Investigation:** Wookyung Park.

**Supervision:** Young-Gun Park.

**Validation:** Mirinae Kim, Rae Young Kim, Young-Gun Park, Young-Hoon Park.

**Visualization:** Young-Hoon Park.

**Writing – original draft:** Wookyung Park.

**Writing – review & editing:** Wookyung Park, Mirinae Kim, Rae Young Kim, Young-Hoon Park.

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
