## [Decision Letter · Decision Letter 0]

15 Jun 2020

PONE-D-20-14747

Long-term Visual Prognosis and Characteristics of Recurrent Retinal Detachment after Silicone Oil removal

PLOS ONE

Dear Dr. Park,

Thank you for submitting your manuscript to PLOS ONE. After careful consideration, we feel that it has merit but does not fully meet PLOS ONE’s publication criteria as it currently stands. Therefore, we invite you to submit a revised version of the manuscript that addresses the points raised during the review process.

Both reviewers found the study interesting and made comments to improve it. We will look forward to the revised version 

We look forward to receiving your revised manuscript.

Kind regards,

Demetrios G. Vavvas

Academic Editor

PLOS ONE

Journal Requirements:

https://bjo.bmj.com/content/bjophthalmol/85/10/1203.full.pdf

In your revision ensure you cite all your sources (including your own works), and quote or rephrase any duplicated text outside the methods section.

Further consideration is dependent on these concerns being addressed.

Reviewers' comments:

Reviewer's Responses to Questions

**Comments to the Author**

1. Is the manuscript technically sound, and do the data support the conclusions?

Reviewer #1: Yes

Reviewer #2: Yes

2. Has the statistical analysis been performed appropriately and rigorously? 

Reviewer #1: I Don't Know

Reviewer #2: I Don't Know

3. Have the authors made all data underlying the findings in their manuscript fully available?

Reviewer #1: No

Reviewer #2: No

4. Is the manuscript presented in an intelligible fashion and written in standard English?

Reviewer #1: No

Reviewer #2: No

5. Review Comments to the Author

Reviewer #1: The authors present a large cohort of RD eyes treated with PPV with silicon oil tamponade.

They report 8.9% re-detachment rate after silicon oil removal in a 50 month follow up and found that this was associated with worse final BCVA compared to the non re-detached group.

Longer period of silicon oil in the eye was associated with worse visual outcome potentially due to higher corneal opacity rates. The authors are advised to consider the following recommendations to eventually strengthen their work:

1. Lines 120-121: ‘The presence of scleral buckling or the number of mean previous retinal surgeries were not different between the primary retinal attachment group and the re-detachment group after removal of SO.’

How would the authors explain their finding that scleral buckle was not associated with lower re-detachment rates following silicon oil removal ? Lines 178-179 do not clearly and adequately elaborate on this.

2. Line 87: Please define ‘’exploratory data analysis’’. What statistical test was used for comparisons between 3 groups ? (<3months,3-6months, >6months)

3. Lines 80-81: Was Swept source Optical Coherence Tomography (OCT) performed in all patients since 2009?

4. Line 82: Please define ‘’and the other unwanted complications development’’

Line 86: Please define stratification of macula status

5. Line 96: Please add a percentage for TRD as with RRD

6. Lines 157-159: Please proof read for the proper use of English. Also, language errors could be found throughout the manuscript. Please proof read the entire work.

7. Please change ‘primary attachment group’ to ‘no re-detachment group’

8. The authors need to explain their finding that the presence of PVR did not increase the chance of re-detachment in this study. Lines 164-168 do not clearly and adequately elaborate on this.

9. Line 167: please define ‘and so on’

10. Limitations should be at the ned of the discussion - not in the middle

11. Abstract: ‘side effects’ should read ‘adverse events’

Abstracts occlusion ‘for too long’ is too vague. Please be more specific in your conclusions.

12. Line 205 ‘The main purpose was to evaluate the risk factors leading to, and to examine the factors indicating, retinal re-detachment.’’ what are the conclusions of the authors ? How do they reply to their main research question ‘what are the risk factors for retinal re-detachment following SO removal?’ It seems there is no clear risk factor per the authors results. Please provide a robust conclusion in the manuscript.

Reviewer #2: This study adds to the literature significant information reporting on the long term visual prognosis and recurrence factors after pars plans vitrectomy and use of silicon oil for the treatment of retinal detachment. The authors in general provided enough data to support their conclusions. Furthermore, the statistical analysis seems to be well conducted. However, I would like a statistician to look at it. The following are a few points that need to be reviewed.

Major points

As far as the language is concerned, a few sentences lack consistency, clarity and syntax (e.g. lines 85-86, 95-97, 143-145, discussion), making it difficult to follow. It is advisable that the authors proofread the text to improve the flow and the readability of the manuscript. If need be, the authors could work with an expert.

The authors should rewrite the introduction of their study and give a more general scope at it. Rhegmatogenous retinal detachment is only one specific type of retinal detachment. Moreover, the other types of retinal detachment included in the study are not mentioned at all, giving the opinion that only RRD eyes participated.

Since a great percentage of the included patients has had at least one retinal surgery in the past, it is important to be clarified in the methods whether the RD was primarily occurred to the eyes studied or there were patients with recurrent RD that were included in the study. Moreover, it might be helpful, if the authors referred to any exclusion criteria they used, concerning the recurrent RDs (e.g. treated in other center and presented in recurrence, other primary treatment option, etc)

The authors should rewrite the discussion aiming at a more comprehensive presentation of previous data. They should revise and clarify a few sentences (e.g. lines 161, 163-166, 173-175, 177-179, 182, 185-186, 227) and provide previous evidence exactly according to references. They should also include references for every percentage or result they report, in order to avoid confusion and render it easier for the reader to relate their findings to previous studies. It is advisable that they firstly report the outcomes of their study and secondly refer and relate to previous findings, in order to avoid possible confusion of the presented outcomes and a “back and forth” analysis.

An additional possible limitation of the study could be the fact that there was statistically significant difference in the three groups of different SO tamponade period, as far as BCVA before SO injection is concerned (table 4). The authors should consider how this difference could influence the last F/U findings and report it in their study (lines 221-223).

In the last paragraph of the manuscript, the authors refer to data that are not available in the study. The authors should make the whole data they used fully available for the reader, especially when they want to emphasize on a result arose from them.

Minor points

Line 78: It should be clarified whether the BCVA was obtained at least six months after the vitrectomy or the SO removal (line 32).

Table 2: It is depicted that a total of 370 eyes underwent concomitant scleral bucking surgery, whereas in Table 1 and in text it is reported that a total of 369 eyes underwent this procedure concomitantly.

Lines 212-213: Τhe groups of this specific analysis should also be mentioned.

I would like to look at a revised version of the manuscript.

6. PLOS authors have the option to publish the peer review history of their article (what does this mean?). If published, this will include your full peer review and any attached files.

Reviewer #1: No

Reviewer #2: No

---

## [Author Response · Author response to Decision Letter 0]

3 Jan 2021

Response to reviewers

Review Comments to the Author

Reviewer #1: The authors present a large cohort of RD eyes treated with PPV with silicon oil tamponade.

They report 8.9% re-detachment rate after silicon oil removal in a 50 month follow up and found that this was associated with worse final BCVA compared to the non re-detached group.

Longer period of silicon oil in the eye was associated with worse visual outcome potentially due to higher corneal opacity rates. The authors are advised to consider the following recommendations to eventually strengthen their work:

1. Lines 120-121: ‘The presence of scleral buckling or the number of mean previous retinal surgeries were not different between the primary retinal attachment group and the re-detachment group after removal of SO.’

How would the authors explain their finding that scleral buckle was not associated with lower re-detachment rates following silicon oil removal ? Lines 178-179 do not clearly and adequately elaborate on this.

In our study, the difference in the recurrence rate cannot be explained simply by the presence or absence of a buckle. However, if the PVR was removed well as planned, it would have produced good results without recurrence even if the buckle was not accompanied by sufficiently removing the inflammatory cells. Follow-up studies on matched control will be helpful.

2. Line 87: Please define ‘’exploratory data analysis’’. What statistical test was used for comparisons between 3 groups ? (<3months,3-6months, >6months)

Thanks for the details. To compare the three groups, ANOVA test and Kruskal-Wallis test were performed on continuous variables, and Chi-squared test and Fisher's exact test were performed on categorical variables.

3. Lines 80-81: Was Swept source Optical Coherence Tomography (OCT) performed in all patients since 2009?

Thanks for the sharp point. SS-OCT was introduced to the institution in 2013. SD-OCT and SS-OCT are used together in this study.

4. Line 82: Please define ‘’and the other unwanted complications development’’

Line 86: Please define stratification of macula status

Line 82: Thanks. Noted that unwanted complications include secondary glaucoma, corneal opacity, and pththisis bulbi.

Line 86: It was to confirm whether retina was re-detached through OCT, but the meaning was expressed vaguely. We deleted the ambiguous part.

5. Line 96: Please add a percentage for TRD as with RRD

The percentage for TRD is also presented. Thank you.

6. Lines 157-159: Please proof read for the proper use of English. Also, language errors could be found throughout the manuscript. Please proof read the entire work.

Thanks for the point. We have corrected the parts you mentioned and other vaguely expressed sentences.

7. Please change ‘primary attachment group’ to ‘no re-detachment group’

Thanks. As you mentioned, ’primary attachment group’ was modified to ‘no re-detachment group’. 

8. The authors need to explain their finding that the presence of PVR did not increase the chance of re-detachment in this study. Lines 164-168 do not clearly and adequately elaborate on this.

Modified and supplemented. Thanks for the point.

9. Line 167: please define ‘and so on’

Thank you for pointing out the point. In addition to the contents expressed, whether the band has been raised well and if the SRF has been sufficiently removed would have influenced the results. 

10. Limitations should be at the end of the discussion - not in the middle

Thanks for the details. Paragraphs related to limitations were sent to the back.

11. Abstract: ‘side effects’ should read ‘adverse events’

Abstracts occlusion ‘for too long’ is too vague. Please be more specific in your conclusions.

Thanks for the sharp point. Modified to ‘adverse events’. Ambiguous expressions have also been corrected.

12. Line 205 ‘The main purpose was to evaluate the risk factors leading to, and to examine the factors indicating, retinal re-detachment.’’ what are the conclusions of the authors ? How do they reply to their main research question ‘what are the risk factors for retinal re-detachment following SO removal?’ It seems there is no clear risk factor per the authors results. Please provide a robust conclusion in the manuscript.

Thank you. Other factors did not show any significant difference, but it was confirmed that the duration of SO tamponade had a difference in visual acuity after surgery and did not significantly affect retinal detachment. If it is determined that the retina is sufficiently stable, it is not necessary to delay the timing of SO removal.

Reviewer #2: This study adds to the literature significant information reporting on the long term visual prognosis and recurrence factors after pars plans vitrectomy and use of silicon oil for the treatment of retinal detachment. The authors in general provided enough data to support their conclusions. Furthermore, the statistical analysis seems to be well conducted. However, I would like a statistician to look at it. The following are a few points that need to be reviewed.

Major points

As far as the language is concerned, a few sentences lack consistency, clarity and syntax (e.g. lines 85-86, 95-97, 143-145, discussion), making it difficult to follow. It is advisable that the authors proofread the text to improve the flow and the readability of the manuscript. If need be, the authors could work with an expert.

The authors should rewrite the introduction of their study and give a more general scope at it. Rhegmatogenous retinal detachment is only one specific type of retinal detachment. Moreover, the other types of retinal detachment included in the study are not mentioned at all, giving the opinion that only RRD eyes participated.

Thanks for the sharp point. As pointed out, we modified the introduction section to include other types of RD.

Since a great percentage of the included patients has had at least one retinal surgery in the past, it is important to be clarified in the methods whether the RD was primarily occurred to the eyes studied or there were patients with recurrent RD that were included in the study. Moreover, it might be helpful, if the authors referred to any exclusion criteria they used, concerning the recurrent RDs (e.g. treated in other center and presented in recurrence, other primary treatment option, etc)

Thanks for your attention to detail. We mentioned in the methods that the surgery also included cases that had recurred after previous surgery. No matter where the previous surgery was done, all cases of RD were included in the study, so exclusion criteria were not specifically addressed for recurrence.

The authors should rewrite the discussion aiming at a more comprehensive presentation of previous data. They should revise and clarify a few sentences (e.g. lines 161, 163-166, 173-175, 177-179, 182, 185-186, 227) and provide previous evidence exactly according to references. They should also include references for every percentage or result they report, in order to avoid confusion and render it easier for the reader to relate their findings to previous studies. It is advisable that they firstly report the outcomes of their study and secondly refer and relate to previous findings, in order to avoid possible confusion of the presented outcomes and a “back and forth” analysis.

Thank you for the clear and accurate point. I have refined and corrected the parts you said.

An additional possible limitation of the study could be the fact that there was statistically significant difference in the three groups of different SO tamponade period, as far as BCVA before SO injection is concerned (table 4). The authors should consider how this difference could influence the last F/U findings and report it in their study (lines 221-223).

Thanks for the good point. At first glance, it can be thought that the worse the preoperative vision was, the worse the preoperative retinal condition was anatomically or functionally. It is difficult to give a clear explanation of how the surgery before SO injection influenced the final difference in visual acuity because there is a limitation of a table that compares groups by dividing the group only by tamponade period without considering other variables thoroughly. Matching all of those variables will lead to more accurate conclusions in subsequent studies.

In the last paragraph of the manuscript, the authors refer to data that are not available in the study. The authors should make the whole data they used fully available for the reader, especially when they want to emphasize on a result arose from them.

The sentence was corrected using only the figures presented. Thanks.

Minor points

Line 78: It should be clarified whether the BCVA was obtained at least six months after the vitrectomy or the SO removal (line 32).

Thanks. Corrected the sentence more clearly.

Table 2: It is depicted that a total of 370 eyes underwent concomitant scleral bucking surgery, whereas in Table 1 and in text it is reported that a total of 369 eyes underwent this procedure concomitantly.

There seems to be an error in the count. Has been modified. The concomitant buckling was done at 369eyes.

Lines 212-213: Τhe groups of this specific analysis should also be mentioned.

Thanks for the detailed point. The text has been revised to clarify the meaning.

---

## [Decision Letter · Decision Letter 1]

9 Feb 2021

PONE-D-20-14747R1

Long-term Visual Prognosis and Characteristics of Recurrent Retinal Detachment after Silicone Oil removal

PLOS ONE

Dear Dr. Park,

Thank you for submitting your manuscript to PLOS ONE. After careful consideration, we feel that it has merit but does not fully meet PLOS ONE’s publication criteria as it currently stands. Therefore, we invite you to submit a revised version of the manuscript that addresses the points raised during the review process.

We look forward to receiving your revised manuscript.

Kind regards,

Demetrios G. Vavvas

Academic Editor

PLOS ONE

Reviewers' comments:

Reviewer's Responses to Questions

**Comments to the Author**

1. If the authors have adequately addressed your comments raised in a previous round of review and you feel that this manuscript is now acceptable for publication, you may indicate that here to bypass the “Comments to the Author” section, enter your conflict of interest statement in the “Confidential to Editor” section, and submit your "Accept" recommendation.

Reviewer #1: (No Response)

Reviewer #2: (No Response)

2. Is the manuscript technically sound, and do the data support the conclusions?

Reviewer #1: Partly

Reviewer #2: Yes

3. Has the statistical analysis been performed appropriately and rigorously? 

Reviewer #1: Yes

Reviewer #2: Yes

4. Have the authors made all data underlying the findings in their manuscript fully available?

Reviewer #1: Yes

Reviewer #2: Yes

5. Is the manuscript presented in an intelligible fashion and written in standard English?

Reviewer #1: Yes

Reviewer #2: No

6. Review Comments to the Author

Reviewer #1: The abstract should be able to stand alone. That means the factors mentioned in methods, should be also mentioned in results and thats where the conclusions should derive from. These sections should match.

In the abstract's methods the authors mention ‘Best-corrected visual acuity was obtained preoperatively and at least 6 months after the SO removal.’ yet pre-op and 6 months post-op BCVA is nowhere in the abstract’s results section.

Conversely, subgroups are mentioned in the results section that were never defined in the methods section. BCVAs need to be added in the abstract at least in parentheses. (for example after the phrase ‘The longest tamponade group showed the worst final vision after SO removal’). The ‘presence of accompanying buckles’ ‘ the number of operations’ and ‘systemic diseases’ are mentioned in methods as factors to be investigated yet are nowhere in the abstract’s results section.

Abstract conclusions: ‘not for too long’ is vague. What are the key conclusions/take home messages for the reader ?

The structure of the main manuscript’s discussion is still confusing. As reviewer #2 pointed out in the previous round of review ‘It is advisable that they firstly report the outcomes of their study and secondly refer and relate to previous findings, in order to avoid possible confusion of the presented outcomes and a “back and forth” analysis.’ This does not seem to have changed, the reader is struggling to identify the results of this study in a compact manner within this discussion and the flow is not easy to follow. Please revise accordingly.

Minor points

Lines 82-83 should read: Either Swept source Optical Coherence Tomography (OCT) (Topcon, Japan) or spectral-domain OCT (Heidelberg, Germany), since I assume the study eyes did not get both tests and as mentioned by the authors in their point-by-point response SS-OCT was introduced in their clinic in 2013.

Line 113, 286: Please change ‘primary attachment group’ to ‘no re-detachment group’

Reviewer #2: Although the revised version has rendered the manuscript of a higher quality, a few of my previous comments were not fully addressed. It is advisable that you take into consideration the following points and revise the manuscript accordingly.

1) Concerning the language of the text, a few sentences continue to lack clarity and syntax (e.g. lines in the revised paper 89-90, 99-101, 147-150). A careful proofreading of the whole text is suggested to improve the language and the flow of the manuscript. Please take into consideration the specific lines mentioned.

2) Lines 50-52: References should be included. What about the exudative type of RD?

3) Line 81: "Best-corrected visual acuity (BCVA) was obtained preoperatively and at least 6 months after the vitrectomy.” still contradicts line 32 in abstract “was obtained preoperatively and at least 6 months after the SO removal.” It should be clarified when the BCVA was exactly obtained.

4) Table 2: It still depicts that a total of 370 eyes underwent concomitant scleral bucking surgery, whereas, as authors report, a total of 369 eyes underwent this procedure concomitantly.

5) Line 148: The word “between" had better be replaced with “compared to”.

6)Line 171: Since an outcome of the current study is reported, it is not clear why the reference [6] is included here.

7) Lines 251-255: These results have been discussed again in the previous paragraph.

7. PLOS authors have the option to publish the peer review history of their article (what does this mean?). If published, this will include your full peer review and any attached files.

Reviewer #1: No

Reviewer #2: No

---

## [Author Response · Author response to Decision Letter 1]

26 Mar 2021

Response to Reviewers

Reviewer #1: The abstract should be able to stand alone. That means the factors mentioned in methods, should be also mentioned in results and thats where the conclusions should derive from. These sections should match.

In the abstract's methods the authors mention ‘Best-corrected visual acuity was obtained preoperatively and at least 6 months after the SO removal.’ yet pre-op and 6 months post-op BCVA is nowhere in the abstract’s results section.

Conversely, subgroups are mentioned in the results section that were never defined in the methods section. BCVAs need to be added in the abstract at least in parentheses. (for example after the phrase ‘The longest tamponade group showed the worst final vision after SO removal’). The ‘presence of accompanying buckles’ ‘ the number of operations’ and ‘systemic diseases’ are mentioned in methods as factors to be investigated yet are nowhere in the abstract’s results section.

Abstract conclusions: ‘not for too long’ is vague. What are the key conclusions/take home messages for the reader ?

Thans for the advice. As you pointed, the abstract was summarized and refined, and the conclusion part was revised more clearly.

The structure of the main manuscript’s discussion is still confusing. As reviewer #2 pointed out in the previous round of review ‘It is advisable that they firstly report the outcomes of their study and secondly refer and relate to previous findings, in order to avoid possible confusion of the presented outcomes and a “back and forth” analysis.’ This does not seem to have changed, the reader is struggling to identify the results of this study in a compact manner within this discussion and the flow is not easy to follow. Please revise accordingly.

Thanks for the point. The structure of the discussion section has been modified to make it easier to understand.

Minor points

Lines 82-83 should read: Either Swept source Optical Coherence Tomography (OCT) (Topcon, Japan) or spectral-domain OCT (Heidelberg, Germany), since I assume the study eyes did not get both tests and as mentioned by the authors in their point-by-point response SS-OCT was introduced in their clinic in 2013.

Modified and supplemented. Thanks for the point.

Line 113, 286: Please change ‘primary attachment group’ to ‘no re-detachment group’

Thanks for the details. It is corrected.

Reviewer #2: Although the revised version has rendered the manuscript of a higher quality, a few of my previous comments were not fully addressed. It is advisable that you take into consideration the following points and revise the manuscript accordingly.

1) Concerning the language of the text, a few sentences continue to lack clarity and syntax (e.g. lines in the revised paper 89-90, 99-101, 147-150). A careful proofreading of the whole text is suggested to improve the language and the flow of the manuscript. Please take into consideration the specific lines mentioned.

Thanks for the point. We have corrected the parts you mentioned and other vaguely expressed sentences.

2) Lines 50-52: References should be included. What about the exudative type of RD?

Thanks. We inserted a reference. The exudative type was not included as an indication for surgery.

3) Line 81: "Best-corrected visual acuity (BCVA) was obtained preoperatively and at least 6 months after the vitrectomy.” still contradicts line 32 in abstract “was obtained preoperatively and at least 6 months after the SO removal.” It should be clarified when the BCVA was exactly obtained.

The sentence was corrected to say that BCVA was measured once before surgery and again after 6 months of operation. Thank you.

4) Table 2: It still depicts that a total of 370 eyes underwent concomitant scleral bucking surgery, whereas, as authors report, a total of 369 eyes underwent this procedure concomitantly.

Thanks for the point. We have corrected an error.

5) Line 148: The word “between" had better be replaced with “compared to”.

Thanks for the details. Correction was made.

6)Line 171: Since an outcome of the current study is reported, it is not clear why the reference [6] is included here.

Thanks for pointing out. Removed unnecessary reference mark.

7) Lines 251-255: These results have been discussed again in the previous paragraph.

Thanks for the sharp point. Removed duplicate content.

---

## [Decision Letter · Decision Letter 2]

28 Apr 2021

PONE-D-20-14747R2

Long-term Visual Prognosis and Characteristics of Recurrent Retinal Detachment after Silicone Oil removal

PLOS ONE

Dear Dr. Park,

Thank you for submitting your manuscript to PLOS ONE. After careful consideration, we feel that it has merit but does not fully meet PLOS ONE’s publication criteria as it currently stands. Therefore, we invite you to submit a revised version of the manuscript that addresses the points raised during the review process.

The study has improved substantially but the reviewers still find some minor areas that need improvement. Please follow their advice and submit the fully revised version

We look forward to receiving your revised manuscript.

Kind regards,

Demetrios G. Vavvas

Academic Editor

PLOS ONE

Journal Requirements:

Reviewers' comments:

Reviewer's Responses to Questions

**Comments to the Author**

1. If the authors have adequately addressed your comments raised in a previous round of review and you feel that this manuscript is now acceptable for publication, you may indicate that here to bypass the “Comments to the Author” section, enter your conflict of interest statement in the “Confidential to Editor” section, and submit your "Accept" recommendation.

Reviewer #1: (No Response)

Reviewer #2: (No Response)

2. Is the manuscript technically sound, and do the data support the conclusions?

Reviewer #1: Yes

Reviewer #2: Yes

3. Has the statistical analysis been performed appropriately and rigorously? 

Reviewer #1: Yes

Reviewer #2: Yes

4. Have the authors made all data underlying the findings in their manuscript fully available?

Reviewer #1: Yes

Reviewer #2: Yes

5. Is the manuscript presented in an intelligible fashion and written in standard English?

Reviewer #1: Yes

Reviewer #2: No

6. Review Comments to the Author

Reviewer #1: Abstract: The phrase ''if the retina is structurally stable enough, it is advisable to consider SO removal'' doe not derive from the authors results. What is '' structurally stable enough'' and how was it evaluated ?

Exclusion criteria: on line 78 need to be more detailed. All pre-existing diseases affecting the macula or optic nerve were excluded ? How about trace ERM ?

Methods line 84: ‘’Best-corrected visual acuity (BCVA) was measured before surgery and 6 months after the vitrectomy.’’ Was BCVA not evaluated in the cases that the silicon oil removal happened more than 6 months after vitrectomy ? Was BCVA not evaluated in the last follow up visit that you say it was a mean of 56 months ? Please be consistent. Methods should say we will evaluate A, B , C, D and results should MATCH those A, B, C, D. Describe in the methods all the comparisons that you intend to present in your results

Conclusion is too general. What are the take home messages to the reader? what were the key findings in this study ? When should we remove SO and which factors show high risk of re-detachment? Answer the questions raised by your title.

Reviewer #2: My previous comments were not fully addressed. The authors should take into consideration the following points.

Abstract:

Comparisons between no re-detachment and detachment groups were also conducted by the authors, though not referred in the abstract, neither in the methods nor the results sections. The authors should provide an abstract that reflects the whole methodology of their study and the investigations they conducted. They should revise the afore mentioned sections accordingly.

Line 39: “The average BCVA was 1.60 ± 0.75 before SO removal and 1.29 ± 0.96 after the removal.”: it should be mentioned that these results refer to LogMAR.

In addition, preoperative BCVA measurements, i.e. before SO injection, are still not provided in results section of the abstract, as reviewer #1 pointed out in the previous round of review. They should be added.

Moreover, as far as the abstract is concerned, P values of statistical significance should also be provided.

Patients and Methods

The authors responded to my previous comment regarding the exudative type of RD that “The exudative type was not included as an indication for surgery.” On the contrary, in methods section (lines 76-77) they report that RD cases such as RRD, tractional retinal detachment (TRD) and proliferative vitreoretinopathy (PVR) were included. In addition to that, Table 1 does not include all indications for surgery, since the sum of the percentages do not equal 100%. Were there other indications and which? The authors should clarify whether exudative type of RD was also investigated and revise the provided table with demographics in order to present complete data.

As I also mentioned in both previous round of reviews, it still remains unclear and confusing, when the BCVA was exactly obtained. More specifically, “after the SO removal” (as it is referred in abstract, results and discussion) means that BCVA was obtained, after the SO tamponade period had been completed, which differs among the studied eyes. In contrast, in line 84, it is mentioned that “Best-corrected visual acuity (BCVA) was measured ….6 months after the vitrectomy.”, which implies that BCVA was measured 6 months after the vitrectomy, independently of the SO tamponade period and SO removal. These two constitute two different time points. The authors should clarify when the BCVA was exactly obtained. It is advisable that abstract, methods, results and provided tables match regarding this information. Additionally, in line 116 the authors refer to the same time point using the term: “measured at the latest visit”. They had also better refer to a specific time point with the same term each time in order to avoid any misunderstanding.

Discussion

Line 172: “Patients with PVR and RD had relapses in 14.8% after SO removal”. This percentage is not correct. The percentage of patients with RD and present PVR (n=158) that had re-detachment after SO removal (n=13) is 8% (13/158=0.08), which differs a lot from the provided percentage, i.e. 14.8%. The authors should revise accordingly.

Lines 193-194: “which did not show a statistically significant difference in the RD recurrence rate between the group with and without the buckle band.”: the authors should modify how they present the results of their study, so that the latter absolutely reflect the precise analyses they conducted. Thus, the previous sentence had better be replaced with: “the percentage of concomitant buckling did not differ significantly between the “no re-detachment” and “re-detachment” groups.” Furthermore, this comparison concerns all types of RD and not only RD with PVR, as the referenced findings of Jost et al. The authors should revise accordingly.

Lines 228-243: This paragraph remains confusing. The authors refer to results of a lot different analyses, i.e. at first they refer to the comparison between the group with reduced or stationary vision and the group with improved vision after SO removal, then to the analysis for each period of SO tamponade concerning visual acuity, corneal opacity and RD recurrence and except for that they also mention a possible limitation of their study, as far as BCVA is concerned. They should rewrite the paragraph and present the outcomes in a more accurate, sound and appropriate manner. They should also consider referring to these outcomes at the beginning of discussion section and not at the end, as pointed out in previous rounds of review, too.

Lines 221-225 and lines 230-232 report the same results. My previous comment regarding these lines was not addressed. Please revise them.

Although significantly different BCVAs before SO injection were referred in discussion section, they were not presented as a possible limitation of the present study. The authors should make a comment in this section.

Minor points

Since there are a lot of grammatical and language mistakes in the manuscript, a careful proofreading of the whole text is still needed in order to render it sound and clear. Some of my following points address precisely lines that should be revised.

Lines 31-32: “Best-corrected visual acuity (BCVA) was before surgery and 6 months after the SO removal.”: the word “obtained” should be added.

Lines 93-94: The Mann–Whitney U test was used to compare the predefined outcome measures the SO-treated eyes.: this sentence lacks syntax and clarity. Please revise it.

Line 102: “was the second most cause of surgery”: the word “frequent” should be added.

Lines 112, 115, 129-130: “primary retinal attachment group” should be replaced with “no re-detachment group”.

Line 188: the word “was” should be changed to “is”.

Line 206: it should be changed to “in contrast to”.

Lines 209-213: “Probably in most areas of clinical surgical field today, the same equipment, instruments and similar surgical techniques were used, and the surgical experience and results of five surgeons in this study were very similar. In a further study, we also could find out the difference from a surgeon factor meanwhile environmental condition should be maintained the same.”: it is advisable that the authors revise these sentences as far as the syntax is concerned, in order to render clear enough what they imply.

Line 219: “But as our knowledge,..”: this phrase should be replaced.

Line 224: “but showed a significant difference”: the sentence lacks subject.

Line 225: “were not arose” should be changed to “did not arise”.

Line 225-226: “especially much higher in long term follow up group.”: the phrase should be revised.

Line 235: “but since there is no adjustment for variables other than the tamponade period. It cannot be concluded.”: this should be one sentence.

Lines 263: Replace “suggest” with suggests

7. PLOS authors have the option to publish the peer review history of their article (what does this mean?). If published, this will include your full peer review and any attached files.

Reviewer #1: No

Reviewer #2: No

---

## [Author Response · Author response to Decision Letter 2]

15 Jul 2021

Reviewer #1: 

Abstract: The phrase ''if the retina is structurally stable enough, it is advisable to consider SO removal'' doe not derive from the authors results. What is '' structurally stable enough'' and how was it evaluated ?

Thanks for the advice. Ambiguous sentences have been corrected to convey the meaning more clearly.

Exclusion criteria: on line 78 need to be more detailed. All pre-existing diseases affecting the macula or optic nerve were excluded ? How about trace ERM ?

It has been explained in more detail.

Methods line 84: ‘’Best-corrected visual acuity (BCVA) was measured before surgery and 6 months after the vitrectomy.’’ Was BCVA not evaluated in the cases that the silicon oil removal happened more than 6 months after vitrectomy ? Was BCVA not evaluated in the last follow up visit that you say it was a mean of 56 months ? Please be consistent. Methods should say we will evaluate A, B , C, D and results should MATCH those A, B, C, D. Describe in the methods all the comparisons that you intend to present in your results

Thanks for the point. Ambiguous expressions have been corrected.

Conclusion is too general. What are the take home messages to the reader? what were the key findings in this study ? When should we remove SO and which factors show high risk of re-detachment? Answer the questions raised by your title.

The conclusion is presented more clearly. Thank you very much for your review.

Reviewer #2:

My previous comments were not fully addressed. The authors should take into consideration the following points.

Abstract:

Comparisons between no re-detachment and detachment groups were also conducted by the authors, though not referred in the abstract, neither in the methods nor the results sections. The authors should provide an abstract that reflects the whole methodology of their study and the investigations they conducted. They should revise the afore mentioned sections accordingly.

Thanks for the point. The descriptions were also added to the corresponding sessions.

Line 39: “The average BCVA was 1.60 ± 0.75 before SO removal and 1.29 ± 0.96 after the removal.”: it should be mentioned that these results refer to LogMAR.

Thanks for the advice. The ‘logMAR’ term was added.

In addition, preoperative BCVA measurements, i.e. before SO injection, are still not provided in results section of the abstract, as reviewer #1 pointed out in the previous round of review. They should be added.

Preoperative BCVA measurement data was added.

Moreover, as far as the abstract is concerned, P values of statistical significance should also be provided.

Patients and Methods

The authors responded to my previous comment regarding the exudative type of RD that “The exudative type was not included as an indication for surgery.” On the contrary, in methods section (lines 76-77) they report that RD cases such as RRD, tractional retinal detachment (TRD) and proliferative vitreoretinopathy (PVR) were included. In addition to that, Table 1 does not include all indications for surgery, since the sum of the percentages do not equal 100%. Were there other indications and which? The authors should clarify whether exudative type of RD was also investigated and revise the provided table with demographics in order to present complete data.

Thanks for the advice. During data analysis, traumatic RD was separately classified, and while organizing the data in the table, this part was omitted. Traumatic RD patients were re-examined, and it was confirmed that all the patients had tear or dialysis. So all of them were reclassified into the RRD group.

As I also mentioned in both previous round of reviews, it still remains unclear and confusing, when the BCVA was exactly obtained. More specifically, “after the SO removal” (as it is referred in abstract, results and discussion) means that BCVA was obtained, after the SO tamponade period had been completed, which differs among the studied eyes. In contrast, in line 84, it is mentioned that “Best-corrected visual acuity (BCVA) was measured ….6 months after the vitrectomy.”, which implies that BCVA was measured 6 months after the vitrectomy, independently of the SO tamponade period and SO removal. These two constitute two different time points. The authors should clarify when the BCVA was exactly obtained. It is advisable that abstract, methods, results and provided tables match regarding this information. Additionally, in line 116 the authors refer to the same time point using the term: “measured at the latest visit”. They had also better refer to a specific time point with the same term each time in order to avoid any misunderstanding.

Thanks for the sharp point. BCVA was measured before SO inj surgery, 1 month, before SO removal, 1, 3, 6 months, and 1 year, and also at the last visit. The visual acuity at the last visit used for data analysis was the last visual acuity tested annually after surgery.

Discussion

Line 172: “Patients with PVR and RD had relapses in 14.8% after SO removal”. This percentage is not correct. The percentage of patients with RD and present PVR (n=158) that had re-detachment after SO removal (n=13) is 8% (13/158=0.08), which differs a lot from the provided percentage, i.e. 14.8%. The authors should revise accordingly.

There seems to be some confusion when analyzing the data. Thanks for pointing out.

Lines 193-194: “which did not show a statistically significant difference in the RD recurrence rate between the group with and without the buckle band.”: the authors should modify how they present the results of their study, so that the latter absolutely reflect the precise analyses they conducted. Thus, the previous sentence had better be replaced with: “the percentage of concomitant buckling did not differ significantly between the “no re-detachment” and “re-detachment” groups.” Furthermore, this comparison concerns all types of RD and not only RD with PVR, as the referenced findings of Jost et al. The authors should revise accordingly.

Thank you. The contents of this study have been modified to be delivered more smoothly.

Lines 228-243: This paragraph remains confusing. The authors refer to results of a lot different analyses, i.e. at first they refer to the comparison between the group with reduced or stationary vision and the group with improved vision after SO removal, then to the analysis for each period of SO tamponade concerning visual acuity, corneal opacity and RD recurrence and except for that they also mention a possible limitation of their study, as far as BCVA is concerned. They should rewrite the paragraph and present the outcomes in a more accurate, sound and appropriate manner. They should also consider referring to these outcomes at the beginning of discussion section and not at the end, as pointed out in previous rounds of review, too.

We have edited the paragraph to be more accurate and appropriate. Thank you.

Lines 221-225 and lines 230-232 report the same results. My previous comment regarding these lines was not addressed. Please revise them.

The discussion section has been corrected to be more appropriate. Thank you.

Although significantly different BCVAs before SO injection were referred in discussion section, they were not presented as a possible limitation of the present study. The authors should make a comment in this section.

Thank you. The points you mentioned were specified as the limitations of this study.

Minor points

Since there are a lot of grammatical and language mistakes in the manuscript, a careful proofreading of the whole text is still needed in order to render it sound and clear. Some of my following points address precisely lines that should be revised.

Lines 31-32: “Best-corrected visual acuity (BCVA) was before surgery and 6 months after the SO removal.”: the word “obtained” should be added.

Thanks for the advice. As you pointed, the sentence was corrected.

Lines 93-94: The Mann–Whitney U test was used to compare the predefined outcome measures the SO-treated eyes.: this sentence lacks syntax and clarity. Please revise it.

The sentence was revised.

Line 102: “was the second most cause of surgery”: the word “frequent” should be added.

The word was added.

Lines 112, 115, 129-130: “primary retinal attachment group” should be replaced with “no re-detachment group”.

The sentences were corrected.

Line 188: the word “was” should be changed to “is”.

The sentence was corrected.

Line 206: it should be changed to “in contrast to”.

The sentence was revised.

Lines 209-213: “Probably in most areas of clinical surgical field today, the same equipment, instruments and similar surgical techniques were used, and the surgical experience and results of five surgeons in this study were very similar. In a further study, we also could find out the difference from a surgeon factor meanwhile environmental condition should be maintained the same.”: it is advisable that the authors revise these sentences as far as the syntax is concerned, in order to render clear enough what they imply.

Line 219: “But as our knowledge,..”: this phrase should be replaced.

The sentence has been corrected.

Line 224: “but showed a significant difference”: the sentence lacks subject.

The sentence was corrected.

Line 225: “were not arose” should be changed to “did not arise”.

The sentence was revised.

Line 225-226: “especially much higher in long term follow up group.”: the phrase should be revised.

The sentence has been corrected.

Line 235: “but since there is no adjustment for variables other than the tamponade period. It cannot be concluded.”: this should be one sentence.

The correction has been made.

Lines 263: Replace “suggest” with suggests

Thanks for the advice. The word was corrected.

---

## [Decision Letter · Decision Letter 3]

12 Aug 2021

PONE-D-20-14747R3

Long-term Visual Prognosis and Characteristics of Recurrent Retinal Detachment after Silicone Oil removal

PLOS ONE

Dear Dr. Park,

Thank you for submitting your manuscript to PLOS ONE. After careful consideration, we feel that it has merit but does not fully meet PLOS ONE’s publication criteria as it currently stands. Therefore, we invite you to submit a revised version of the manuscript that addresses the points raised during the review process.

both reviewers found the MS improved but with still some areas in need for improvement 

We look forward to receiving your revised manuscript.

Kind regards,

Demetrios G. Vavvas

Academic Editor

PLOS ONE

Journal Requirements:

Reviewers' comments:

Reviewer's Responses to Questions

**Comments to the Author**

1. If the authors have adequately addressed your comments raised in a previous round of review and you feel that this manuscript is now acceptable for publication, you may indicate that here to bypass the “Comments to the Author” section, enter your conflict of interest statement in the “Confidential to Editor” section, and submit your "Accept" recommendation.

Reviewer #1: (No Response)

Reviewer #2: (No Response)

2. Is the manuscript technically sound, and do the data support the conclusions?

Reviewer #1: Partly

Reviewer #2: Yes

3. Has the statistical analysis been performed appropriately and rigorously? 

Reviewer #1: I Don't Know

Reviewer #2: Yes

4. Have the authors made all data underlying the findings in their manuscript fully available?

Reviewer #1: Yes

Reviewer #2: Yes

5. Is the manuscript presented in an intelligible fashion and written in standard English?

Reviewer #1: No

Reviewer #2: Yes

6. Review Comments to the Author

Reviewer #1: Please mention specific page and lines that the changes are made, as well as the phrase that was added or changed in quotation marks for each one of the reviewers comments. Otherwise the reviewers cannot track the changes that you have or haven't made. Please avoid general phrases such as ‘’Ambiguous expressions have been corrected.’’ Be specific in your point-by-point response. Additional comments:

Abstract Methods: the phrase: After SO removal, the anatomic outcomes were compared between the re-detached group and the no re-detached group.’ needs correction. The groups mentioned were not previously defined. Anatomical outcomes between re-detached and not re-detached retinas ? These are be definition different.

Abstract Methods: Subgroup analysis looking for what outcome ?

Abstract Methods: other complications is too general. Either be specific or omit from the abstract.

Abstract Results: ‘’Overall, visual acuity improved after removal of oil than before surgery.’’ P-value is missing. Was this statistically significant ?

Abstract Results: ‘’There was no significant difference’’ and you show a p value of <0.001 ?? Clarify your sentence if you mean after So removal VA significantly different between the 2 groups.

Abstract Results: ‘’than the other two groups with shorter tamponade periods’’ what are the 2 other groups ??

Line 99: what are the 3 groups ? This has not been defined.

Please write a conclusion that answers the research questions that your study intended to answer based on your results.

Reviewer #2: There are a few points that still need to be revised.

Abstract - Lines 45-47: P values of statistical significance should be provided here, too.

Lines 88-89: “BCVA was measured before SO injection surgery and 1 month after the surgery, before SO removal surgery, 1, 3, 6 months, and 1 year after removal.”: The authors should not use commas if it is not necessary, because it can lead to misconceptions. Do “1, 3, 6 months” refer to the period before or after SO removal? In the period before SO removal, was BCVA evaluated only one month after SO injection surgery or at additional time points, e.g. 1, 3 and 6 months? These points remain confusing and need to be clarified.

If the evaluation of BCVA before SO removal was only conducted one month after the SO injection surgery, independently from the SO tamponade period of each group, it should be mentioned that whenever “Before SO removal” is used in the provided tables and results, it refers to that specific time point, i.e. one month after SO injection surgery. It is advisable that the authors mention this correspondence.

Lines 88-90: Since the time point of last follow up differed among patients, it should be included as a possible limitation of the present study. BCVA at last follow up would have been more appropriately comparable between groups, if the last follow up referred to a specific time point for all patients, e.g. 6 months after SO removal. Other factors could possibly have affected BCVA within longer follow up periods.

Line 98: “measures" should be replaced with measurements.

Lines 215-217: As I have already commented, this comparison of the study concerns all types of RD and not only RD with PVR. This difference should be mentioned and rendered clear enough to the reader, since all other studies, that are referenced at this paragraph, investigated RD with PVR cases. The authors could first refer to their outcomes (lines 215-217) and then present the results of other studies, that were focused precisely on RD with PVR.

Furthermore, Deaner et al and Garweg et al reported specifically outcomes about eyes with retinal detachment complicated by proliferative vitreoretinopathy, a fact that should also be reported.

Lines 243-244: It should be replaced with: “An additional limitation of this study is the fact that visual acuity before SO injection was significantly different in the three groups of different SO tamponade period and this could have affected the visual prognosis after surgery.”

7. PLOS authors have the option to publish the peer review history of their article (what does this mean?). If published, this will include your full peer review and any attached files.

Reviewer #1: No

Reviewer #2: No

---

## [Author Response · Author response to Decision Letter 3]

9 Feb 2022

Response to Reviewers

Reviewer #1: 

Abstract: The phrase ''if the retina is structurally stable enough, it is advisable to consider SO removal'' doe not derive from the authors results. What is '' structurally stable enough'' and how was it evaluated ?

Thanks for the advice. Ambiguous sentences have been corrected to convey the meaning more clearly.

Exclusion criteria: on line 78 need to be more detailed. All pre-existing diseases affecting the macula or optic nerve were excluded ? How about trace ERM ?

It has been explained in more detail.

Methods line 84: ‘’Best-corrected visual acuity (BCVA) was measured before surgery and 6 months after the vitrectomy.’’ Was BCVA not evaluated in the cases that the silicon oil removal happened more than 6 months after vitrectomy ? Was BCVA not evaluated in the last follow up visit that you say it was a mean of 56 months ? Please be consistent. Methods should say we will evaluate A, B , C, D and results should MATCH those A, B, C, D. Describe in the methods all the comparisons that you intend to present in your results

Thanks for the point. Ambiguous expressions have been corrected.

Conclusion is too general. What are the take home messages to the reader? what were the key findings in this study ? When should we remove SO and which factors show high risk of re-detachment? Answer the questions raised by your title.

The conclusion is presented more clearly. Thank you very much for your review.

Reviewer #2:

My previous comments were not fully addressed. The authors should take into consideration the following points.

Abstract:

Comparisons between no re-detachment and detachment groups were also conducted by the authors, though not referred in the abstract, neither in the methods nor the results sections. The authors should provide an abstract that reflects the whole methodology of their study and the investigations they conducted. They should revise the afore mentioned sections accordingly.

Thanks for the point. The descriptions were also added to the corresponding sessions.

Line 39: “The average BCVA was 1.60 ± 0.75 before SO removal and 1.29 ± 0.96 after the removal.”: it should be mentioned that these results refer to LogMAR.

Thanks for the advice. The ‘logMAR’ term was added.

In addition, preoperative BCVA measurements, i.e. before SO injection, are still not provided in results section of the abstract, as reviewer #1 pointed out in the previous round of review. They should be added.

Preoperative BCVA measurement data was added.

Moreover, as far as the abstract is concerned, P values of statistical significance should also be provided.

Patients and Methods

The authors responded to my previous comment regarding the exudative type of RD that “The exudative type was not included as an indication for surgery.” On the contrary, in methods section (lines 76-77) they report that RD cases such as RRD, tractional retinal detachment (TRD) and proliferative vitreoretinopathy (PVR) were included. In addition to that, Table 1 does not include all indications for surgery, since the sum of the percentages do not equal 100%. Were there other indications and which? The authors should clarify whether exudative type of RD was also investigated and revise the provided table with demographics in order to present complete data.

Thanks for the advice. During data analysis, traumatic RD was separately classified, and while organizing the data in the table, this part was omitted. Traumatic RD patients were re-examined, and it was confirmed that all the patients had tear or dialysis. So all of them were reclassified into the RRD group.

As I also mentioned in both previous round of reviews, it still remains unclear and confusing, when the BCVA was exactly obtained. More specifically, “after the SO removal” (as it is referred in abstract, results and discussion) means that BCVA was obtained, after the SO tamponade period had been completed, which differs among the studied eyes. In contrast, in line 84, it is mentioned that “Best-corrected visual acuity (BCVA) was measured ….6 months after the vitrectomy.”, which implies that BCVA was measured 6 months after the vitrectomy, independently of the SO tamponade period and SO removal. These two constitute two different time points. The authors should clarify when the BCVA was exactly obtained. It is advisable that abstract, methods, results and provided tables match regarding this information. Additionally, in line 116 the authors refer to the same time point using the term: “measured at the latest visit”. They had also better refer to a specific time point with the same term each time in order to avoid any misunderstanding.

Thanks for the sharp point. BCVA was measured before SO inj surgery, 1 month, before SO removal, 1, 3, 6 months, and 1 year, and also at the last visit. The visual acuity at the last visit used for data analysis was the last visual acuity tested annually after surgery.

Discussion

Line 172: “Patients with PVR and RD had relapses in 14.8% after SO removal”. This percentage is not correct. The percentage of patients with RD and present PVR (n=158) that had re-detachment after SO removal (n=13) is 8% (13/158=0.08), which differs a lot from the provided percentage, i.e. 14.8%. The authors should revise accordingly.

There seems to be some confusion when analyzing the data. Thanks for pointing out.

Lines 193-194: “which did not show a statistically significant difference in the RD recurrence rate between the group with and without the buckle band.”: the authors should modify how they present the results of their study, so that the latter absolutely reflect the precise analyses they conducted. Thus, the previous sentence had better be replaced with: “the percentage of concomitant buckling did not differ significantly between the “no re-detachment” and “re-detachment” groups.” Furthermore, this comparison concerns all types of RD and not only RD with PVR, as the referenced findings of Jost et al. The authors should revise accordingly.

Thank you. The contents of this study have been modified to be delivered more smoothly.

Lines 228-243: This paragraph remains confusing. The authors refer to results of a lot different analyses, i.e. at first they refer to the comparison between the group with reduced or stationary vision and the group with improved vision after SO removal, then to the analysis for each period of SO tamponade concerning visual acuity, corneal opacity and RD recurrence and except for that they also mention a possible limitation of their study, as far as BCVA is concerned. They should rewrite the paragraph and present the outcomes in a more accurate, sound and appropriate manner. They should also consider referring to these outcomes at the beginning of discussion section and not at the end, as pointed out in previous rounds of review, too.

We have edited the paragraph to be more accurate and appropriate. Thank you.

Lines 221-225 and lines 230-232 report the same results. My previous comment regarding these lines was not addressed. Please revise them.

The discussion section has been corrected to be more appropriate. Thank you.

Although significantly different BCVAs before SO injection were referred in discussion section, they were not presented as a possible limitation of the present study. The authors should make a comment in this section.

Thank you. The points you mentioned were specified as the limitations of this study.

Minor points

Since there are a lot of grammatical and language mistakes in the manuscript, a careful proofreading of the whole text is still needed in order to render it sound and clear. Some of my following points address precisely lines that should be revised.

Lines 31-32: “Best-corrected visual acuity (BCVA) was before surgery and 6 months after the SO removal.”: the word “obtained” should be added.

Thanks for the advice. As you pointed, the sentence was corrected.

Lines 93-94: The Mann–Whitney U test was used to compare the predefined outcome measures the SO-treated eyes.: this sentence lacks syntax and clarity. Please revise it.

The sentence was revised.

Line 102: “was the second most cause of surgery”: the word “frequent” should be added.

The word was added.

Lines 112, 115, 129-130: “primary retinal attachment group” should be replaced with “no re-detachment group”.

The sentences were corrected.

Line 188: the word “was” should be changed to “is”.

The sentence was corrected.

Line 206: it should be changed to “in contrast to”.

The sentence was revised.

Lines 209-213: “Probably in most areas of clinical surgical field today, the same equipment, instruments and similar surgical techniques were used, and the surgical experience and results of five surgeons in this study were very similar. In a further study, we also could find out the difference from a surgeon factor meanwhile environmental condition should be maintained the same.”: it is advisable that the authors revise these sentences as far as the syntax is concerned, in order to render clear enough what they imply.

Line 219: “But as our knowledge,..”: this phrase should be replaced.

The sentence has been corrected.

Line 224: “but showed a significant difference”: the sentence lacks subject.

The sentence was corrected.

Line 225: “were not arose” should be changed to “did not arise”.

The sentence was revised.

Line 225-226: “especially much higher in long term follow up group.”: the phrase should be revised.

The sentence has been corrected.

Line 235: “but since there is no adjustment for variables other than the tamponade period. It cannot be concluded.”: this should be one sentence.

The correction has been made.

Lines 263: Replace “suggest” with suggests

Thanks for the advice. The word was corrected.

Reviewer #1: Please mention specific page and lines that the changes are made, as well as the phrase that was added or changed in quotation marks for each one of the reviewers comments. Otherwise the reviewers cannot track the changes that you have or haven't made. Please avoid general phrases such as ‘’Ambiguous expressions have been corrected.’’ Be specific in your point-by-point response. Additional comments:

Abstract Methods: the phrase: After SO removal, the anatomic outcomes were compared between the re-detached group and the no re-detached group.’ needs correction. The groups mentioned were not previously defined. Anatomical outcomes between re-detached and not re-detached retinas ? These are be definition different.

Page 2, line 33

The sentence was corrected to ‘The anatomical results were compared between the group in which the retina was detached again after SO removal and the group in which the retina was not detached.’.

Abstract Methods: Subgroup analysis looking for what outcome ?

Page 2, line 36

It was indicated that subgroup analysis was performed to determine whether the duration of SO tamponade had an effect on RD recurrence.

Abstract Methods: other complications is too general. Either be specific or omit from the abstract.

Page 2, line 38

‘other complications’ were deleted.

Abstract Results: ‘’Overall, visual acuity improved after removal of oil than before surgery.’’ P-value is missing. Was this statistically significant ?

Page 2, line 44

That sentence has been deleted.

Abstract Results: ‘’There was no significant difference’’ and you show a p value of <0.001 ?? Clarify your sentence if you mean after So removal VA significantly different between the 2 groups.

Page 2, line 46

Corrected the sentence with ‘but visual acuity of re-detachment group was worse than no re-detachment group after SO removal (p<0.001)’.

Abstract Results: ‘’than the other two groups with shorter tamponade periods’’ what are the 2 other groups ??

Page 3, line 49

They are the groups with SO tamponade duration of less than 3 months and between 3 and 6 months.

Line 99: what are the 3 groups ? This has not been defined.

Page 5, line 102

They are the groups divided by the duration of the SO tamponade. Description was added.

Please write a conclusion that answers the research questions that your study intended to answer based on your results.

Reviewer #2: There are a few points that still need to be revised.

Abstract - Lines 45-47: P values of statistical significance should be provided here, too.

Page 3, line 51

P values were added.

Lines 88-89: “BCVA was measured before SO injection surgery and 1 month after the surgery, before SO removal surgery, 1, 3, 6 months, and 1 year after removal.”: The authors should not use commas if it is not necessary, because it can lead to misconceptions. Do “1, 3, 6 months” refer to the period before or after SO removal? In the period before SO removal, was BCVA evaluated only one month after SO injection surgery or at additional time points, e.g. 1, 3 and 6 months? These points remain confusing and need to be clarified.

If the evaluation of BCVA before SO removal was only conducted one month after the SO injection surgery, independently from the SO tamponade period of each group, it should be mentioned that whenever “Before SO removal” is used in the provided tables and results, it refers to that specific time point, i.e. one month after SO injection surgery. It is advisable that the authors mention this correspondence.

Page 4, line 93

BCVA was measured when hospitalized for SO removal surgery, and was measured at 1 month, 3 months, 6 months, and 1 year after removal, respectively. The manuscript was revised to reflect these contents.

Lines 88-90: Since the time point of last follow up differed among patients, it should be included as a possible limitation of the present study. BCVA at last follow up would have been more appropriately comparable between groups, if the last follow up referred to a specific time point for all patients, e.g. 6 months after SO removal. Other factors could possibly have affected BCVA within longer follow up periods.

Page 13, line 253

The points you figured out were additionally described as limitations of the study. Thank you.

Line 98: “measures" should be replaced with measurements.

Page 5, line 104

Thanks for the comment. As you said, I modified it with measurements.

Lines 215-217: As I have already commented, this comparison of the study concerns all types of RD and not only RD with PVR. This difference should be mentioned and rendered clear enough to the reader, since all other studies, that are referenced at this paragraph, investigated RD with PVR cases. The authors could first refer to their outcomes (lines 215-217) and then present the results of other studies, that were focused precisely on RD with PVR.

Furthermore, Deaner et al and Garweg et al reported specifically outcomes about eyes with retinal detachment complicated by proliferative vitreoretinopathy, a fact that should also be reported.

Lines 243-244: It should be replaced with: “An additional limitation of this study is the fact that visual acuity before SO injection was significantly different in the three groups of different SO tamponade period and this could have affected the visual prognosis after surgery.”

Page 13, line 250

Edited as you mentioned. thank you.

---

## [Editor Report · Decision Letter 4]

28 Feb 2022

Long-term Visual Prognosis and Characteristics of Recurrent Retinal Detachment after Silicone Oil removal

PONE-D-20-14747R4

Dear Dr. Park,

We’re pleased to inform you that your manuscript has been judged scientifically suitable for publication and will be formally accepted for publication once it meets all outstanding technical requirements.

Kind regards,

Demetrios G. Vavvas

Academic Editor

PLOS ONE
---

## [Editor Report · Acceptance letter]

6 Apr 2022

PONE-D-20-14747R4 

Long-term Visual Prognosis and Characteristics of Recurrent Retinal Detachment after Silicone Oil removal 

Dear Dr. Park:

I'm pleased to inform you that your manuscript has been deemed suitable for publication in PLOS ONE. Congratulations! Your manuscript is now with our production department. 

Kind regards, 

on behalf of

Prof. Demetrios G. Vavvas 

Academic Editor

PLOS ONE